# Retinoids in Fungal Infections: From Bench to Bedside

**DOI:** 10.3390/ph14100962

**Published:** 2021-09-24

**Authors:** Terenzio Cosio, Roberta Gaziano, Guendalina Zuccari, Gaetana Costanza, Sandro Grelli, Paolo Di Francesco, Luca Bianchi, Elena Campione

**Affiliations:** 1Dermatology Unit, Department of Systems Medicine, University of Rome Tor Vergata, Via Montpellier 1, 00133 Rome, Italy; terenziocosio@gmail.com (T.C.); luca.bianchi@uniroma2.it (L.B.); 2Department of Experimental Medicine, University of Rome Tor Vergata, 00133 Rome, Italy; roberta.gaziano@uniroma2.it (R.G.); difra@uniroma2.it (P.D.F.); 3Department of Pharmacy, University of Genoa, Viale Cembrano, 16148 Genoa, Italy; zuccari@difar.unige.it; 4Department of Experimental Medicine, University of Rome Tor Vergata, Via Montpellier 1, 00133 Rome, Italy; costanza@uniroma2.it (G.C.); grelli@med.uniroma2.it (S.G.)

**Keywords:** retinoids, *Candida* spp., *Aspergillus* spp., mycosis, onychomycosis, nanoparticles, *Malassezia* spp., dermatophytes, microbiology, mycology, all-trans retinoic acid

## Abstract

Retinoids—a class of chemical compounds derived from vitamin A or chemically related to it—are used especially in dermatology, oncohematology and infectious diseases. It has been shown that retinoids—from their first generation—exert a potent antimicrobial activity against a wide range of pathogens, including bacteria, fungi and viruses. In this review, we summarize current evidence on retinoids’ efficacy as antifungal agents. Studies were identified by searching electronic databases (MEDLINE, EMBASE, PubMed, Cochrane, Trials.gov) and reference lists of respective articles from 1946 to today. Only articles published in the English language were included. A total of thirty-nine articles were found according to the criteria. In this regard, to date, In vitro and In vivo studies have demonstrated the efficacy of retinoids against a broad-spectrum of human opportunistic fungal pathogens, including yeast fungi that normally colonize the skin and mucosal surfaces of humans such as *Candida* spp., *Rhodotorula mucilaginosa* and *Malassezia furfur*, as well as environmental moulds such as *Aspergillus* spp., *Fonsecae monofora* and many species of dermatophytes associated with fungal infections both in humans and animals. Notwithstanding a lack of double-blind clinical trials, the efficacy, tolerability and safety profile of retinoids have been demonstrated against localized and systemic fungal infections.

## 1. Introduction

In the last 20 years, the incidence of invasive fungal infections (IFI) has increased significantly [1]. From a global perspective, *Candida* spp. and *Aspergillus* spp. represent the most common opportunistic fungal pathogens associated with systemic infections, especially in severely immunocompromised individuals. However, the above are not the only infections, as there are many new cases of fungal diseases linked to environmental opportunistic fungi, which were previously considered non-pathogenic for humans. Among these, *Cryptococcus neoformans* is one of the main causes of morbidity and mortality due to systemic mycosis associated with acquired immunodeficiency syndrome (AIDS), with an estimated 600,000 deaths per year. There are also a wide range of pathogens belonging to the order of Mucorales (e.g., *Rhizopus* spp.), such as ialoifomycetes (*Fusarium* and *Scedosporium* spp.) or pheoifomycetes (*Alternaria* spp. and *Cladophialophora bantiana*), which can be encountered [2]. Mycoses caused by these organisms, especially rhino-cerebral forms, are often difficult to treat and require specialist advice [3]. The high incidence of invasive and non-invasive fungal infections in recent years is due to the increased prevalence of immunocompromised subjects, especially for iatrogenic causes, such as oncological chemotherapies, corticosteroid-based therapies to cure autoimmune diseases, and subjects suffering from AIDS in countries with limited access to treatments [4]. Thanks to improved surgical management and the introduction of selective immunosuppressive drugs, there are also subjects undergoing solid organ or bone marrow transplantation, who are at high risk for the development of fungal infections in the neutropenic phase. Other predisposing conditions are represented by the increased use of intravenous devices, prolonged hospitalization in intensive care units and the administration of antibacterial therapies, which alter the normal human microbiota [4]. The Global Action Fund for Fungal Infections (GAFFI) has calculated that around the world, at least 150 patients die every hour from IFI. This means a total of 1,350,000 deaths a year. It is true to say that some neutropenic cancer patients must undergo antifungal therapy in one-third of cases when they do not respond to broad-spectrum antibiotic therapies, because of their predisposition to IFI [5]. The development of drug-resistant strains and the high toxicity of traditional antifungal drugs, especially in the case of prolonged and extensive use, have increasingly strengthened the need to find new and more effective therapeutic strategies. In particular, the search for new antifungal agents aims at identifying molecules with greater selectivity, less cytotoxicity and less chance of developing drug resistance phenomenon in fungi. There is, therefore, a need to find new effective therapies in the field of fungal infections with a safer profile for the patient and easy to access and manage clinically. Historically, vitamin A was considered as an anti-microbial agent, but the mechanisms associated with its anti-infective properties remain mostly hypothetical and are probably related to its pleiotropic effects on the immune response [6]. All-trans retinoic acid (ATRA), also known as tretinoin, is vitamin A’s more potent naturally occurring derivative. It has been reported that the administration of vitamin A or its metabolite, ATRA, decreases the incidence and severity of infectious diseases, although their modulation of the immune function may also vary widely, depending on the type of infection and the immune responses involved [7,8]. The term “retinoid” concerns both natural and synthetic analogues of vitamin A. The first generation of retinoids comprises natural derivatives (retinol, tretinoin, isotretinoin, and alitretinoin), obtained by modifying the polar end group of vitamin A [9]. The second generation refers to more lipophilic monoaromatic compounds such as etretinate and acitretin, in which a benzene ring replaces the cyclohexene ring. Finally, the third generation is made up by rigid polyaromatic molecules, resulting from the cyclization of the unsaturated side chain (adapalene, bexarotene, and tazarotene). The fixed structure reduces broad interactions, thus increasing drug selectivity. Trifarotene is a recently synthetized fourth-generation retinoid, highly specific for skin RAR-γ receptors [10]. A milestone in the possible application of retinoids as antifungal agents arose after an observation of evidence-based medicine (EBM), gone unnoticed by Girmenia et al. [11] but elucidated by Campione et al. [12]. Girmenia et al. [11] analyzed the incidence and type of infections complicating the clinical course of 89 consecutive acute promyelocytic leukemia (APL) patients receiving the all-trans retinoic acid plus idarubicin (AIDA) protocol. A total of 179 febrile episodes were registered during induction and consolidation, most of them due to coagulase-negative staphylococci and viridans group streptococci, while fungal infections were only occasionally observed. This clinical observation paved the way for the possible use of retinoid in the prevention and treatment of mycosis [11]. Campione et al. [12] reported the In vitro fungistatic effect of tazarotene 0.1% against *Candida albicans* and *Candida glabrata* and the In vivo effect in the treatment of onychomycosis caused by *Trichophyton* spp. [12]. For this observation, they hypothesized if also ATRA, the active form of vitamin A, could be effective in preventing and treating fungal infections, highlighting the results obtained from Girmenia et al. [11]. Moreover, Campione et al. [13] demonstrated the efficacy of ATRA In vitro and In vivo against *Aspergillus niger,* thus suggesting the use of retinoids in mycosis [13]. From these results, retinoids have been investigated as a possible agent in the treatment of mycoses.

The aim of our review is to highlight the current application of retinoids as antimycotic agents due to their direct or indirect antifungal activity, by means of the immunoadjuvant properties of these compounds, opening up new scenarios for the future use of retinoids in the treatment of localized and disseminated mycoses.

## 2. Methods and Study Design

### 2.1. Search Strategy

We performed a comprehensive search in the following databases from 1946 to July 2021: Cochrane Central Register of Controlled Trials; MEDLINE; Embase; US National Institutes of Health Ongoing Trials Register; NIHR Clinical Research Network Portfolio Database; and the World Health Organization International Clinical Trials Registry Platform. We studied reference lists and published systematic review articles. We used the term “retinoid” with the following keywords, separately and in combination: “Fungal Infection”, “fungal biofilm”, “dermatophytoma”, *“Entomophthoromycota”, “Basidiobolus”, “Conidiobolus”, “Ascomycota”, “Ajellomycetaceae”, “Paracoccidioides”, “Lacazia”, “Coccidioides”, “Blastomyces”, “Histoplasma”, “Sporothrix”, “Talaromyces”, “Trichophyton”, “Microsporum”, “Epidermophyton”, “Rhizopus“, “Mucor”, “Malassezia”, “Micrococcus” “Cladophialphora”, “Ramichloridium”, “Exophiala”, “Curvularia”, “Alternaria”, “Fusarium”, “Aspergillus”, “Candida”, “Fonsecaea”, “Rhodotorula”.* Only English language articles were included in the searches. Forward citation searching of the reference lists of the original studies and review articles was also conducted.

### 2.2. Inclusion Criteria

To investigate the direct and indirect effect of retinoids against fungal pathogens, if the study included the retinoids with other drugs, only the retinoid frame was analyzed. All human studies were included, with no restrictions on age, sex, ethnicity or type of study. Case reports and case series were included if they described the use of retinoids in diseases that were not included in reviews or trials.

### 2.3. Exclusion Criteria

The target intervention excluded the analyses of other pathologies not due to fungal pathogens, and non-English language articles.

## 3. Results

Three hundred and ninety-seven articles or trials regarding retinoids and fungal infections were identified by this quantitative research. Three hundred and fifty-six were excluded after the application of exclusion criteria. Among the forty-one articles or trials eligible for evaluation, two were excluded after abstract or full text reading. Thirty-nine articles or trials were evaluated in this review (Appendix A). The results of our research are summarized in Table 1.

### 3.1. Effectiveness of Retinoids against Opportunistic Fungi That Colonize the Skin and Mucosae in Humans

#### 3.1.1. Candida

Infections due to *Candida* spp. are the major causes of morbidity and mortality among hospitalized patients and are associated with a wide variety of clinical manifestations, ranging from superficial and mucosal infections to life-threatening disseminated bloodstream candidemia [14]. Global estimates suggest that invasive candidiasis occurs in more than a quarter of a million patients every year, with incidence rates for candidemia of 2–14 per 100,000 inhabitants in population-based studies [15]. Antifungal drug resistance, including multidrug resistance (MDR) in *Candida* spp., has become increasingly important in the management of invasive fungal infections. These infections are related to high morbidity and mortality rates and can be linked to healthcare-associated transmission. In this view, the finding of novel drugs able to act against MDR *Candida* spp. should be a priority. From 2007, novel retinoid derivatives containing a benzimidazole moiety were synthesized and tested for their antimicrobial activity. Their antimicrobial properties against *Staphylococcus aureus*, including methicillin-resistant *Staphylococcus aureus* strains (MRSA)*, Escherichia coli, Pseudomonas aeruginosa, Enterococcus faecalis, Candida krusei* and *Candida albicans* were evaluated. While some of them exhibited moderate activity against *S. aureus,* including MRSA strains, *E. faecalis, C. krusei and C. albicans*, none of the compounds showed any activity against *E. coli* and *P. aeruginosa* [16]. For *Candida* spp., *Candida krusei* (ATTC 6258) and *Candida albicans* (ATCC 10231), minimum inhibitory concentration (MIC) values of 50 μg/mL were obtained against the fungus for the targeted compounds (Figure 1), comparable with fluconazole for *C. krusei* [16]. Moreover, translational research is increasing our knowledge in fungistatic retinoids’ effects, paving the way for their future use in the treatment of mycosis. As previously reported, Campione et al. reported the fungistatic effect of tazarotene 0.1% In vitro against *Candida albicans* and *Candida glabrata* [12]. Scardina et al. [17] investigated the efficacy of isotretinoin for the treatment of nystatin-resistant oral candidiasis. They evaluated six patients with clinical and mycological diagnosis of Candida stomatitis, treated with 0.18% isotretinoin solution applied twice daily, previously treated with nystatin for 30 days. After one month of retinoid therapy, five patients were negative for *Candida* spp., and only one patient with suspected sicca syndrome was found to have oral candidiasis 15 days after the last administration of isotretinoin. None of the patients had any complaints about the medication. These findings prove that 0.18% isotretinoin, applied twice a day for one month, can suppress nystatin-resistant *Candida* oral infection [17]. To date, the current results about In vitro and In vivo application of retinoid against *Candida* spp. deserve future attention as molecules to control candida infection.

#### 3.1.2. Rhodotorula Mucilaginosa

Human fungal infections by *Rhodotorula* spp., a genus of unicellular environmental pigmented yeasts, have increased in the last decades [18]. In China, they are among the main causes of invasive fungal infections by non-*Candida* yeasts [19] and are considered an emerging pathogen. Martini et al. [20] reported a case of 38-year-old man who had whitish nail changes on all fingers as the only symptom. The condition had developed within a few days and led to dystrophy of the proximal part of the nail plates. Microscopic examination of his nail scrapings demonstrated budding hyphae and the patient—who is a teacher—reported frequent use of a wet sponge. Antifungal therapy was started with oral itraconazole pulse therapy (400 mg day for 1 week), combined with the topical application of 5% amorolfine nail lacquer. Subsequent cultures and molecular typing methods identified *Rhodotorula mucilaginosa* [20]. This environmental yeast was repeatedly isolated despite treatment with itraconazole. As no improvement was achieved and testing of the biological activity of the fungus revealed only marginal keratolytic activity, it was considered as a colonizer of a destroyed nail matrix. Finally, a biopsy of the nail bed confirmed the diagnosis of nail psoriasis, which rapidly responded to treatment with oral acitretin (10 mg/day) and topical calcipotriol/betamethasone cream, leading to a rapid clinical improvement after 8 weeks. Fungal growth in the destroyed nails masked the underlying disease and might have triggered the psoriatic nail reaction [20]. Moreover, Jarros et al. [21] isolated *R. mucilaginosa* from a patient with chronic renal disease (CKD). This opportunistic fungus may represent a high risk of serious infection; thus, a correct identification of the yeast is the main means for an efficient treatment [21]. Additionally, as reported in the literature, *R. mucilaginosa* tends to co-infect the nails together with other fungal pathogens, such as *Candida* or *Trichophyton*, paving the way to understanding whether it can infect nails on its own or needs other fungi to develop and promote a crosstalk in a dermatophytoma nail [22,23]. Ge et al. [22] also reported a case of onychomycosis with the concomitant presence of *Rhodotorula mucilaginosa* and *Candida parapsilosis*, while Idris et al. [23] documented a mixed infection of toenails caused by *Trichosporon asahii* and *Rhodotorula mucilaginosa* [22,23]. These cases underline how environmental yeasts deserve our attention as pathogens in clinical practice and strive us to search for new molecules to prevent fungal infections in susceptible individuals.

#### 3.1.3. Malassezia

*Malassezia* spp. are lipid-dependent basidiomycetous yeasts that inhabit the skin and mucosa of humans and other warm-blooded animals, representing a major component of the skin microbiome. They occur as skin commensals, but are also associated with various skin disorders such as tinea versicolor and blood-stream infections [24]. Handojo et al. [25] were the first to propose the clinical use of retinoids in the treatment of *Malassezia*. They carried out a clinical trial to evaluate the cure rate, the incidence of relapse and the tolerance of retinoic acid 0.05% cream applied topically to 50 patients suffering from tinea versicolor. In this trial, two types of topical retinoic acid were used: retinoic acid 0.05% in vanishing cream and retinoic acid 0.05% in equal parts of propylene glycol and 95% alcohol. The patients entered in this trial randomly and were divided into two groups: 25 patients received the 0.05% cream, applied twice daily, and 25 patients received the 0.05% lotion, applied twice daily. Four patients failed to complete the treatment and were thus excluded from evaluation. The number of patients followed up until the end of the treatment period can be specified as follows: (i) 23 patients (92%) in the group receiving the 0.05% cream; (ii) 23 patients (92%) in the group receiving the 0.05% lotion. The treatment period needed to obtain a favorable result was relatively short, namely: 2 weeks in 33 patients (73.33%) and 3 weeks in 45 patients (97.83%) with a mean of 2.27 weeks. The results of this study show that in vivo, retinoic acid has an anti-fungal action, as proved in this trial by the absence of the causative agent on microscopic examination and by the low incidence of relapse. Another important feature of retinoic acid is its deep penetrative capacity, which makes it possible to reach the deeply seated *Malassezia furfur* and destroy it. As far as we know, this deep penetrative capacity of retinoic acid is not found in any other anti-fungal preparation [25]. As regards the above-mentioned criteria for a successful treatment, there was no significant difference between the lotion and cream groups. It was pointed out that patients suffering from tinea versicolor are predisposed to dermatophyte infections, and that cleanliness of the skin is a simple but essential way of preventing contamination with *Malassezia furfur* [25]. Moreover, Shi et al. [26] studied the effect of adapalene gel versus 2% ketoconazole cream in pityriasis versicolor. A total of 100 patients were enrolled in the study, 80 of which were randomized into two arms: one receiving adapalene gel and the other 2% ketoconazole. Among the 80 patients, 67 fulfilled the study criteria and entered evaluation of the curative effect according to the investigator’s requirements. Of the 67 patients, 35 were included in the adapalene group and 32 in the ketoconazole group. One group was treated with 2% ketoconazole cream topically twice daily for 2 weeks, whereas adapalene gel was used for the other group in a similar fashion. No significant differences in efficacy between the two groups were observed: the adapalene gel group presented clinical resolution in 31/40 (77.5%) patients and a negative mycological test in 30/40 (75%) patients after 4 weeks, whilst the ketoconazole cream 2% treated group presented clinical resolution and a negative test in 28/40 (70%) and 28/40 (70%) patients, respectively. No major side effects were noted in either group. In conclusion, the study underlines the clinical efficacy, security and tolerability of adapalene in pityriasis versicolor [26]. Adapalene can strip the abnormal keratinocytes and normalize the dysfunction of keratinization in keratinocytes and epidermal turnover time in the lesions. Moreover, adapalene can decrease the sebum secretion of sebaceous glands in the skin, and thus topical adapalene gel promotes an environment inhospitable to *Malassezia* yeasts—a disadvantage for the propagation of the yeasts—and leads to the elimination of numerous spores and hyphae together with the abnormal cell layer of keratinocytes. On the other hand, adapalene also has anti-inflammatory activity [27]. Based on its immunomodulatory actions, topical adapalene gel was discovered to effectively reduce inflammation in the lesions of pityriasis versicolor, thus improving the symptoms of the disease [27]. Therefore, although adapalene is not a special antifungal drug, it might counteract the fungal growth by affecting the environment required by *Malassezia* yeasts for their living [26]. Yazıcı et al. [28] reported a case of pityriasis versicolor in a 33-year-old man with lesions recurring for 15 years. Medical history revealed the use of many topical and systemic antifungal agents as well as topical retinoic acids with partial benefits. Only oral isotretinoin therapy at a dose of 20 mg/day (0.4 mg/kg/day) has proven to be effective with clinical resolution of pityriasis [28]. Nowadays, In vivo data support the use of retinoids in the treatment of *Malassezia* spp. human infection. Further In vitro studies are needed to elucidate the direct and indirect effects of retinoids.

### 3.2. Effectiveness of Retinoids against Environmental Human Pathogenic Filamentous Fungi

#### 3.2.1. *Aspergillus* spp.

*Aspergillus* is a natural ubiquitous saprophyte found in air, soil and organic matter. Humans normally inhale the spore form of the fungus [29]. *Aspergillus* species cause a wide spectrum of diseases in humans [30]. Depending on the underlying immune status of the host, *Aspergillus* diseases can be roughly classified into three groups with distinct pathogenetic mechanisms, clinical manifestations, and overlapping features [31]. Moreover, aspergillus infections could arise as clinical syndromes in patients with different immune statuses, including severe asthma with fungal sensitization (SAFS), allergic bronchial pulmonary aspergillosis (ABPA), chronic pulmonary aspergillosis (CPA), invasive pulmonary aspergillosis (IPA) and invasive bronchial aspergillosis (IBA) [31]. Furthermore. *Aspergillus* infections do not affect only the respiratory system, but could also involve different apparatuses, including skin appendages. As previously reported by Girmenia et al. [11], a lower infection rate due to fungal pathogens was observed in neutropenic patients with acute promyelocytic leukemia and treated with the AIDA regimen. Apart from the reported results, there is one noteworthy fact: patients suffering from promyelocytic leukemia treated with ATRA have significantly lower incidences of fungal infections caused by *Candida* spp. and *Aspergillu* spp. [11]. Their data were elucidated and underlined by Campione et al. [12], showing a strong In vitro fungistatic effect exerted by ATRA against *Candida albicans* and *Aspergillus fumigatus*, through the inhibition of both germination and hyphal growth of *Candida* yeasts and *Aspergillus* conidia [12]. The authors also demonstrated that ATRA at sub-optimal doses is able to synergize In vitro with the conventional antifungal amphotericin B and posaconazole to counteract the germination of *Aspergillus* conidia [13]. This synergistic effect is of great importance, as it allows one to reduce the dosage of antifungal drugs, limiting exposure to the toxic effects related to their prolonged and massive use. ATRA has also proven to exert a protective effect against aspergillosis in vivo. In an experimental animal model of IPA, ATRA, administered as prophylaxis, significantly increased the survival rate of the animals, compared to the untreated control animals (60% vs. 20% after 12 days of infection). Interestingly, the antifungal efficacy of ATRA was completely comparable to that obtained with posaconazole, one of the antifungals commonly used in the treatment of IPA. Molecular docking studies suggested that the mechanism underlying the antifungal property of ATRA might be due to its ability to interfere with HSP90 activity by competing with the ATP binding-site of the HSP90 protein. HSP90 is an ATP-dependent chaperonin that plays a crucial role in fungal virulence and pathogenicity. It has been shown that ATRA down-regulates the expression of the HSP90-related gene as well as the mRNA expression levels of *AbaA*, *CrzA* and *WetA* genes involved in the conidial germination process. Intriguingly, ATRA enhances In vitro the phagocytosis of *Aspergillus* conidia by macrophages [13]. It could be hypothesized that the protective effect In vivo of this retinoid might be due not only to a direct fungistatic effect but also to the immunoadjuvant properties of the molecule due to its ability to enhance the innate immune response that play a key role in the clearance of fungal infection. ATRA could, therefore, represent a pleiotropic molecule with a great potential either in monotherapy or in a combination with conventional antimycotic drugs for the development of novel antifungal therapeutic strategies aimed at countering the increasing spread of the drug resistance phenomenon in fungal pathogens. As previously reported, *Aspergillus* infection could also affect skin appendages. Onychomycosis (OM) is a chronic fungal infection of the nail caused by dermatophytes, yeasts, and non-dermatophytes. El-Salam et al. [32] evaluated the efficacy of tazarotene 0.1% gel alone or combined with tioconazole nail paint in the treatment of onychomycosis due to *Aspergillus* spp., including *A, candidius*, *A*. *flavus, A. niger, A. nidulans* and *A. terreus* [32]. Forty patients presenting with onychomycosis underwent full history taking, as well as clinical and nail examination, including a clinical and dermoscopic assessment of severity using the Onychomycosis Severity Index (OSI), KOH test, and fungal culture. A significant treatment response was found in those patients treated with tazarotene 0.1% gel administered in combination with tioconazole, compared to tazarotene 0.1% gel alone (decrease in OSI, dermoscopic features, and mycological clearance). Tazarotene alone has been shown to have an antifungal activity especially against *Aspergillus niger* and *A. flavus*, whereas, when combined with antifungals such as tioconazole, it seems to be more effective, suggesting its potential use as adjuvant for standard systemic or topical antifungal treatments of onychomycosis [32]. Based on the immunomodulatory activity of retinoids and given that the nail unit is an immunological niche, a balance between pro-inflammatory and anti-inflammatory cytokines is paramount for the correct management of onychomycosis and other related inflammatory diseases such as psoriasis [33]. *Aspergillus* spp. are also involved in fungal keratitis, a major cause of corneal ulcers, resulting in significant visual impairment and blindness. Zhao et al. [34] investigated the effect of fenretinide in *Aspergillus fumigatus* keratitis In vivo in a mouse model as well as In vitro in a THP-1-derived macrophage cell line infected with *A. fumigatus.* In both experimental models, the pretreatment with fenretinide contributed to protecting corneal transparency during *A. fumigatus* keratitis in the early phase of the infection by reducing neutrophil recruitment, decreasing myeloperoxidase (MPO) levels and increasing apoptosis. Compared with controls, fenretinide also impaired proinflammatory cytokine interleukin 1 beta (IL-1β) production in response to *A. fumigatus* exposure through the blockade of the lectin-type oxidized LDL receptor 1 (LOX-1) and c-Jun N-terminal kinase (JNK) pathway [34]. All retinoids evaluated in *Aspergillus* spp. infections have demonstrated both clinical and In vitro efficacy, setting the stage for further investigation on the use of retinoids in clinical practice.

#### 3.2.2. *Fonsecaea* spp.

Chromoblastomycosis is a subcutaneous fungal infection caused by dematiaceous fungi that belong to the order *Chaetothyriales* and family *Herpotrichiellaceae,* such as as *Fonsecaea pedrosoi*, *Phialophora verrucosa*, and *Cladophialophora carrionii* [35]. This infection is prevalent in tropical and subtropical areas and has been designated as a neglected tropical disease according to the World Health Organization (WHO) [36]. Chromoblastomycosis infection is difficult to treat, and there are limited therapeutic options, making the characterization of new drugs or approaches to treat this infection urgent. Belda et al. [35] reported two cases of extensive chromoblastomycosis lesions due to Fonsecaea spp., treated with a combination of itraconazole (200 mg/day), acitretin (50 mg/kg), and topical imiquimod, 5 days a week for 4 months. In the fourth month of treatment, both patients showed improvement in the verrucous plates, suggesting that acitretin combined with drugs already used in chromoblastomycosis therapy can decrease the duration of treatment, thus improving the patient’s quality of life [37]. Bao et al. [38] reported a case of chromoblastomycosis caused by *Fonsecaea monophora* in a 60-year-old male carpenter with a 40-year history of psoriasis, from Shandong in northern China. A therapy based on 400 mg/day of itraconazole for 5 weeks did not resolve the infection. However, oral itraconazole (200 mg/day), combined with acitretin (20 mg/day), was more effective, and the lesions resolved completely after 1 month of treatment [38]. The rational use of acitretin in chromoblastomycosis is based on its keratoplastic activity and modulation of hyperkeratosis found in this pathology. However, no data regarding the direct action of acitretin against *Fonsecaea* spp. have been reported. From these clinical studies, it would appear that the combined therapy with itraconazole plus acitretin led to the regression of the chromoblastomycosis caused by *F. monophora*, suggesting a potential synergistic effect between the two compounds. Further In vitro and In vivo studies are necessary to understand their interactive effect when used in association.

#### 3.2.3. Dermatophytes

Dermatophytes are a group of fungi able to invade keratinized tissues such as skin, hair and nails, causing infections in humans and animals, collectively named dermatophytosis. These fungi are classified into three genera, including *Trichophyton, Epidermophyton* and *Microsporum*. The genus *Trichophyton* is characterized morphologically by the development of both smooth-walled macro- and microconidia. Clinical manifestations of *Trichophyton* include “tinea”, also called athlete’s foot, ringworm, jock itch, and similar infections of the nail, beard, skin and scalp [39]. Within the wide range of *Trichophyton* clinical presentations, onychomycosis is the most prevalent nail disease, and it is mainly caused by two dermatophyte species, *Trichophyton rubrum* and *Trichophyton interdigitale*, with a frequency in the range of 80% and 20%, respectively [40]. Campione et al. [14] evaluated 15 patients presented with distal and lateral subungual onychomycosis and treated with topical tazarotene 0.1% gel, once per day for 12 weeks. Ten of these patients presented *T. rubrum*, two *T. mentagrophytes*, one presented *T. tonsurans and* one *Epidermophyton floccosum*. Six patients (40%) reached a mycological cure on target nail samples already after 4 weeks of treatment, while sampled material collected after 12 weeks were negative for infections in all patients. Complete clinical healing and negative cultures were reached in all patients at week 12, with a significant improvement in all clinical parameters of the infected nails [14]. Moreover, Campione et al. [14] evaluated the fungistatic effect of tazarotene 0.1% In vitro against *Trichophyton verrucosum* [14]. This agent also displayed In vitro a dose-dependent inhibitory activity, suggesting its direct antifungal effect on the fungus *Trycophyton verrucosum* [14]. Tazarotene is a synthetic third-generation retinoid derived from vitamin A, which has proved to be beneficial in modulating keratinocyte proliferation and in reducing inflammation [14]. In particular, tazarotene effect is mediated by the up-regulation of the intracellular retinoic acid-binding protein-II expression [41]. This protein transactivates nuclear retinoic acid receptors (RARs), in particular RARβ and RARγ, by binding to specific DNA sequences on their promoter gene regions, thus leading to a reduced proliferation of normal and neoplastic cells and favoring cellular differentiation and apoptosis [41]. The anti-inflammatory activity of tazarotene, due to its immunomodulatory properties, may inhibit the fungal keratinolytic proteases, thus contributing to its efficacy. Indeed, tazarotene is able to impair the release of inflammatory cytokines such as IL-6 and interferon gamma (IFN-γ) from mononuclear leukocytes in vitro, as well as the nitric oxide synthesis in a dose-dependent manner [42]. A limitation of our preliminary results is the absence of a randomized placebo control group [14]. Additionally, Handojo et al. [23] showed that the growth of dermatophytes (*Trichophyton, Epidermophyton* and *Microsporum*) on Sabouraud glucose agar medium (pH adjusted to 5.4–5.6) containing ATRA at a concentration of 0.015% was totally inhibited, as well as the growth of contaminants [23]. On the other hand, the fungal growth was only partially inhibited by a concentration of 0.01% [23]. Despite tazarotene being considered a third generation retinoid, the translational research demonstrated the efficacy of ATRA in dermatophytes infection, both for *Aspergillus* spp. and *Candida* spp. The efficacy of ATRA was demonstrated by Gaziano et al. [43]. Among the numerous components of *Cardiospermum halicacabum L. (C. halicacabum)*, an herbaceous climber belonging to the Sapindaceae family, ATRA can exert a clear dose-dependent fungistatic activity In vitro against *Trichophyton rubrum* [43]. These results are in line with Gaziano et al.’s previous analyses showing that ATRA inhibited In vitro the germination of *Candida albicans* and *Aspergillus fumigatus* in a dose-dependent manner [30]. Furthermore, the heat shock protein 90 (Hsp90) chaperone of microbial pathogens could be a possible therapeutic target for *C. halicacabum*. The potential use of chaperones as molecular drug targets could be considered unattractive due to the similarity of the molecular structure between human and microbial chaperones. However, they exhibit different dependencies on chaperone-dependent pathways and could therefore display differences in the sensitivity of their inhibition [30]. In conclusion, Hsp90 may not be the only possible target for *C. halicacabum*, since multiple plant bioactive compounds may interact with different molecular targets both in a single and multiple intracellular pathways [43]. Both In vitro and In vivo data support the action of retinoids against dermatophytes, and actual formulations such as topical 0.1% tazarotene gel in the treatment of OM deserve further attention.

### 3.3. Efficacy of Retinoids against Pneumocystis

#### Pneumocystis

*Pneumocystis jiroveci* is an unusual fungus of the genus *Pneumocystis* [44]. This fungus is the causative agent of Pneumocystis pneumonia (PcP), a common opportunistic disease in immunocompromised hosts, such as patients with AIDS [45] and those with other predisposing immune deficiencies [46]. Historically, PcP has been reported mainly in immunocompromised patients. Pereira-Díaz et al. [47] conducted an observational, descriptive transversal study that included all patients admitted in Spain with discharge diagnosis of PcP, registered in the National Health System’s Hospital Discharge Records Database of Spain between 2008 and 2012. The results of this first nationwide study in Spain allow a change in the misconception that, after the AIDS pandemic, PcP became an infrequent disease, showing that today it is an emerging problem in immunocompromised patients, including without HIV infection [47]. Lei et al. [48] evaluated the effect of ATRA combined with primaquine in a murine model of pneumocystis infection. Their previous studies of various inflammatory components during PcP found that myeloid-derived suppressor cells (MDSCs) accumulate in the lungs of mice and rats with PcP [48]. ATRA (5 mg/kg/day in 8% DMSO) combined with primaquine (2 mg/kg/day in water) displayed a protective effect against PcP, as the combination of trimethoprim and sulfamethoxazole (TMP, 50 mg/kg/day and SMX, 250 mg/kg/day) induced MDSCs’ differentiation into macrophages, which play a key role in the clearance of PcP by recognition, phagocytosis, and degradation of *Pneumocystis* [48,49,50]. Further studies are needed to evaluate the potential combination of retinoids with standard therapy for *P. jiroveci* infection, also considering it as an emerging problem in immunocompromised patients without HIV infection due to immunosuppressive therapies.

**Table 1 pharmaceuticals-14-00962-t001:** Retinoid-based treatments against ungual pathogens.

Fungi	Spp.	Clinical/Experimental Model	Pathological Model	Retinoid	Combination	Results	Reference
*Candida*	*albicans*	In vitro culture		Retinoid derivatives containing a benzimidazole moiety		Antimicrobial activity MIC 1.56μg/mL	[16]
	*albicans*	In vitro culture		1 g of Tazarotene 0.1% gel dissolved in 3mL of physiological solution		Fungistatic activity	[12]
	*glabrata*	In vitro culture		1 g of Tazarotene 0.1% gel dissolved in 3mL of physiological solution		Fungistatic activity	[12]
	*krusei*	In vitro culture		Retinoid derivatives containing a benzimidazole moiety		Antimicrobial activity MIC 1.56μg/mL	[16]
	Not specified	Human	Chronic hyperplastic candidiasis nystatin-resistant	0.18% isotretinoin applied twice a day for one month		Clinical resolution after one month	[17]
*Malassezia*	*furfur*	Human	Pityriasis versicolor	Retinoic acid 0.05% cream vs. retinoic acid 0.05% lotion twice daily for 3 weeks		50 patients totally. 33 patients (73.33%) and 3 weeks in 45 patients (97.83%) with a mean of 2.27 weeks	[25]
	Not specified	Human	Pityriasis versicolor	Adapalene gel vs. ketoconazole 2%		Clinical resolution after 4 weeks and 30/40 (75%) presented mycological negative test vs. Ketoconazole cream 2% 28/40 (70%) and 28/40 (70%)	[26]
	Not specified	Human	Pityriasis versicolor	Oral isotretinoin 20 mg/day (0,4 mg/kg/day) for 6 weeks		Clinical resolution after 6 weeks	[28]
*Aspergillus*	*fumigatus*	structural bioinformatic analysis		ATRA		Competitive inhbitor of the Hsp90 ATP-binding site	[13]
	*fumigatus*	In vitro colture		ATRA		Fungistatic activity (0.5 and 1 mM) down-regulation of HSP90 mRNa and protein expression; enhances the phagocytosis of macrophages (5 or 10mM)	[13]
	*fumigatus*	Rat	Invasive pulmonary aspergillosis (IPA)	ATRA, 2 mg/kg i.p. for 6 days	Alone; versus posaconazole; versus vehicol	Reduction in mortality of IPA	[13]
	*niger*	Human	Onychomycosis	Tazarotene 0.1% gel twice daily for three months	Alone or plus tioconazole (28% nail paint)	Clinical resolution of OM after three months	[32]
	*flavus*	Human	Onychomycosis	Tazarotene 0.1% gel twice daily for three months	Alone or plus tioconazole (28% nail paint)	Clinical resolution of OM after three months	[32]
	*fumigatus*	Murine model	Keratitis	Fenretinide 100 μM subconjunctival injection		Inhibition of neutrophil recruitment and IL-1β production	[34]
*Dermatophyte (Trichophyton)*	*rubrum*	Human	Onychomycosis	Tazarotene 0.1% gel once daily for 12 weeks		Clinical resolution	[12]
	*mentagrophytes*	Human	Onychomycosis	Tazarotene 0.1% gel once daily for 12 weeks		Clinical resolution	[12]
	*verrucosum*	In vitro culture		1 g of Tazarotene 0.1% gel dissolved in 3 mL of physiological solution		Fungistatic activity	[12]
	*tonsurans*	In vitro culture		1 g of Tazarotene 0.1% gel dissolved in 3 mL of physiological solution		Fungistatic activity	[12]
*Dermatophyte (Epidermophyton)*	*floccosum*	Human	Onychomycosis	Tazarotene 0.1% gel once daily for 12 weeks		Fungistatic activity	[12]
*Pneumocystis*	*jiroveci*	Mice and rats	pneumonia	ATRA 5 mg/kg/day in 8% DMSO	Primaquine 2 mg/kg/day in water	Engaged myeloid-derived suppressor cells	[48]
*Fonsecaea*	Not specified	Human	Chromoblastomycosis	Acitretin 50 mg/day	Itraconazole and 200 mg/day topical imiquimod for 5 weeks	Clinical resolution	[37]
	Not specified	Human	Chromoblastomycosis	Acitretin 20 mg/day for 5 weeks	Itraconazole and 200 mg/day topical imiquimod for 5 weeks	Clinical resolution	[37]
	*monophora*	Human	Chromoblastomycosis	Acitretin 20 mg/Kg for 1 month	Itraconazole 200 mg/day for 1 month	Clinical resolution	[38]
*Rhodotorula*	*mucilaginosa*	Human	Onychomycosis and psoriasis	Oral acitretin 10 mg/day for 8 weeks	Topical calcipotriol/betamethasone for 8 weeks	Clinical improvement within 8 weeks	[21]

## 4. Discussion

To date, In vitro and In vivo studies demonstrated a remarkable antifungal efficacy of retinoids against a broad spectrum of opportunistic fungi, as summarized in Table 2.

In vitro studies have demonstrated the efficacy of ATRA against *Candida albicans*, *Aspergillus fumigatus* and dermatophytes; isotretinoin against *Candida albicans*, *Aspergillus fumigatus* and *Aspergillus niger*; and tazarotene against *Candida albicans*, *Candida glabrata* and *Trychophyton verrucosum*. In vivo studies have reported the clinical efficacy of ATRA against *Malassezia furfur* and *Aspergillus fumigatus*; isotretinoin against *Pneumocystis jiroveci* and *Malassezia furfur*; adapalene against *Malassezia furfur*; fenretinide against *Aspergillus fumigatus*; acitretin against *Fonsecaea monophora* and *Rhodotorula mucilaginosa*; and tazarotene against the dermatophytes *Trychophyton rubrum*, *Trychophyton mentagrophytes*, *Trychophyton tonsurans* and *Epidermophyton floccosum*. The chemical structures of the tested compounds are reported in Figure 2. Notwithstanding the In vivo and In vitro efficacy of retinoids, just one trial has evaluated retinoids in fungal infections (Appendix A).

### Unmet Needs in Fungal Infections

Existing anti-fungal agents have a variety of limitations, from toxicity to rising levels of resistance in common fungal pathogens, significant drug interactions and sub-optimal efficacy [51]. The rising proportion of the community in Western countries with induced immune deficiencies caused by immunomodulatory therapies and their expanding indications is gradually increasing the burden of fungal diseases. Equally, unlike anti-bacterial therapies, which have exceptionally good response rates in sensitive pathogens, antifungal therapies are not as efficacious, due to poor host immunity, innate fungal properties and drug dosing limitations [52]. Moreover, drugs administered by the enteral or parenteral routes need to reach the blood circulation to be distributed to all apparatus and act on the infected site. To contemplate retinoids in the prevention and treatment of fungal infections, their common adverse events—such as local erythema, liver toxicity, dry skin, irritation, photosensitization and teratogenic risk—must be considered, as they can reduce the therapeutic adherence to a great extent [53]. Furthermore, retinoids’ efficacy is strongly limited by several disadvantages, including low water solubility, short lifetime due to the degradation by the cytochrome P450-dependent monooxygenase system, and high sensibility to oxygen, light, and heat. Taken together, all these characteristics drastically reduce bioavailability and, consequently, the therapeutic potential [54]. Consequently, a strategy to reduce side-effects related to free retinoid administration, while increasing their bioavailability and maximizing their therapeutic effects, can be represented by the design and development of advanced formulations. With this aim, several drug delivery systems (DDSs), both local and systemic, loaded with retinoids have been developed. DDSs are carriers able to encapsulate active molecules and deliver them to cell/tissue targets in order to reduce the side effects and achieve an increased therapeutic efficacy compared to formulations containing the drug only in the free form. Among retinoids, all-*trans*-retinoic acid (ATRA) represents the first-choice treatment for several skin diseases, and could be used as a future choice in clinical practice for fungal infections. To avoid the aforementioned adverse effects, recently, Zuccari et al. [55] prepared ATRA-loaded micelles (ATRA-TPGSs), by its encapsulation in D-α-tocopheryl-polyethylene-glycol-succinate (TPGS). Loaded micelles of 12 nm mean diameter were further embedded in a Carbopol^®^-based gel and applied on porcine skin for topical release evaluation. The nanogel was capable of improving drug absorption through the stratum corneum by an increment of drug concentration in the formulation without the use of organic solvents such as ethanol and propylene glycol. Their new formulation, characterized by low polydispersity, slightly negative zeta potential, and good encapsulation efficiency, highlights the improvement in patient compliance necessary to achieve better therapeutic outcomes [55]. Fungal keratitis is an infection of the cornea caused by fungi as *Fusarium, Aspergillus* and *Candida* spp. [56]. Although ATRA has proven to be effective against *Candida* and *Aspegillus* spp., the role of retinoids in ophthalmology is controversial. In the 1980s, ocular surface disease emerged as a potential therapeutic target, since vitamin A deficiency was known to cause epithelial squamous metaplasia and glandular atrophy [56]. To date, no new retinoid ophthalmic formulations for fungal keratitis have been developed. This limitation paves the way for future ophthalmic formulation associated with drugs in the guidelines for ocular infections potentially sensitive to ATRA. As previously reported, pulmonary aspergillosis is a collective term used to refer to a number of conditions caused by Aspergillus spp. infections. Moreover, pulmonary drug delivery offers the advantage of positioning the bioactive molecule in direct contact with the pathological lung epithelia, thus ensuring a rapid onset of the therapeutic response. High local drug concentrations may be easily obtained by pulmonary administration with a concomitant increase in the pharmacological effect but without the side effects elicited by other administration routes [57]. As previously reported, fenretinide has shown In vivo activity against *A. fumigatus* infection. This semisynthetic retinoid shows a more favorable toxicological profile, characterized by minimal systemic toxicity and good tolerability. Unfortunately, the drug’s hydrophobic character strongly hinders its aqueous solubilization. Aqueous fenretinide formulations have been obtained by complexation with cyclodextrins [58], encapsulation into nanomicelles [59], or liposomes [60]. Complexation with 2-hydroxypropyl beta-cyclodextrin has increased fenretinide aqueous solubility from 0.017 mg/mL (pure drug) to 2.41 mg/mL (complex). The aqueous formulation of the complexed drug, administered by the parenteral route, was well-tolerated and increased the drug’s bioavailability and antitumor activity in mouse models of different tumor types [58]. Nanoencapsulation in phosphatidylcholine-glyceryltributyrate nanomicelles has increased the fenretinide aqueous solubilization up to 3.88 mg/mL (nanoencapsulated drug). The intravenous administration of the nanomicelles in mice bearing tumor xenografts showed enhanced drug bioavailability and antitumor activity [59]. Moreover, high tolerability was demonstrated by the absence of adverse effects after repeated administrations and for long periods. The ability of liposomes to encapsulate both hydrophilic and lipophilic molecules and to be linked to targeting moieties makes them very promising candidates for drug delivery [60]. Due to their biocompatibility, biodegradability and no immunogenicity, liposomes are the first DDSs that have been translated to clinical applications [61]. Therefore, the In vivo tolerability and the ability to provide high fenretinide solubilization levels suggest that complexation with cyclodextrins [58], encapsulation in nanomicelles [59] or liposomes [60] can be a valuable means for the preparation of safe and efficient pulmonary fenretinide formulations. It has been demonstrated that the encapsulation of various lipophilic nutraceuticals/pharmaceuticals/cosmeceuticals inside lipid-based nanocarrier systems can protect from photo/chemical degradation, improve the aqueous solubility, and allow deeper skin penetration of similar active ingredients. Solid lipid nanoparticles (SLNs) [62], nanostructured lipid carriers (NLCs) [63], liposomes [64], niosomes [65], and nanoemulsions (NEs) [66] are examples of lipid-based systems that have been proven to decrease drug degradation, improve drug targeting, and enhance the efficacy of retinoids in the treatment of skin disorders. Among lipid-based DDSs, ATRA-loaded SLNs are currently one of the most studied DDSs, thanks to their scalability, small particle size, good drug protection, and high drug-loading capacity. Despite these improvements, the drug could undergo expulsion and crystallization during storage. To overcome this drawback, NCLs were performed. Nevertheless, up to now, among the various lipid-based formulations designed to deliver retinoid compounds, NE-based drug delivery systems have been identified as the most feasible and economical method of topical therapy for various skin disorders. Through research advancements using homolipids and heterolipids as excipients, NE formulations have gained much attention due to their ability to enhance the topical efficacy of otherwise poorly permeable retinoid compounds. NEs have demonstrated wide compatibility with different retinoid compounds, surfactants, and oil systems. They are also easy to process and manufacture. This has generated further interest in NEs as drug carriers in the development of various topical formulations [66].

## 5. Conclusions and Future Perspectives

Retinoids have found widespread use in clinical practice, and in recent years, the focus has shifted to their possible antifungal action, both in humans and in experimental In vitro and In vivo models. However, to date, there is a lack of clinical trials that have evaluated the efficacy, tolerability and safety of retinoids against localized or systemic fungal infections. Data from clinical studies show the therapeutic efficacy of retinoids in the treatment of superficial mycoses. In detail, tretinoin and isotretinoin have proven to be effective against *M. furfur*, while tazarotene seems to be the best therapeutic choice for onychomycosis due to dermatophytes. Moreover, tretinoin displays a remarkable antifungal activity both in In vitro and In vivo experimental models against *A. fumigatus* and *C. albicans*, which represent the major opportunistic fungi associated with invasive and life-threatening mycoses in severely immunocompromised patients. Currently, *Candida* biofilm-related infections represent a serious global health threat and are often particularly difficult to treat due to the increasing emergence of multidrug-resistant (MDR) fungal strains. As with bacteria, fungal biofilms are well structured communities in which microorganisms are embedded within self-produced matrix of extracellular polymeric substances (EPSs), composed mainly of polysaccharides, proteins DNA and lipids [67]. Candida is able to produce biofilms on both biotic and abiotic surfaces, such as human tissues during infections and implanted medical devices [68]. Fungal biofilms play a crucial role in the pathogenesis of superficial mucosal and life-threatening systemic mycosis [69]. Embedded within biofilms, the fungal cells are more resistant to antifungal drugs and host immune defense mechanisms than their planktonic counterparts. Therefore, microorganisms which grow in biofilms are able to persist inside the host, causing severe infections, especially in nosocomial settings [70]. Thus, there is an urgent need to develop novel therapeutic strategies to prevent biofilm formation and/or disrupt pre-formed biofilms. In this context, retinoids could represent a novel approach to prevent and reduce biofilm formation. Nowadays, although it has been shown that these agents can exert bactericidal and anti-biofilm activity in bacterial cultures, there are no data about their potential use to counteract biofilm-based fungal infections [71]. In this regard, due to its own fungistatic effect, we hypothesize that ATRA may have great potential against *Candida albicans* biofilm formation by blocking both hyphal extension and budding replication of the yeast cells. Indeed, the hyphal germination is a crucial event in the biofilm formation, contributing to its architectural stability. Furthermore, the fungal cells must reach a critical density to release extracellular molecules that work as auto-inducers to produce an extracellular biofilm matrix. This mechanism, known as quorum sensing (QS), represents another critical step in the development of biofilms. ATRA, by inhibiting the replication of the fungal cells, might also interfere with QS mechanism, thus leading to impaired biofilm formation [72]. Therefore, the potential employment of this molecule could be particularly beneficial in a clinical setting of selected patients, such as individuals undergoing medical devices or organ solid and bone marrow transplantation to prevent systemic mycoses. From this perspective, further studies are needed to assess the potential anti-biofilm activity of this drug. In conclusion, retinoids, either alone or in combination with currently antifungal drugs, could be very promising agents for novel therapeutic or preventive antifungal options, thus overcoming the major clinical hurdle of drug resistance in fungi.

## Figures and Tables

**Figure 1 pharmaceuticals-14-00962-f001:**
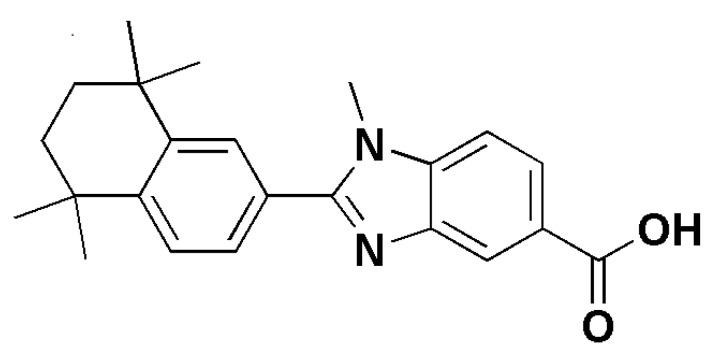
1-methyl-2-(5,5,8,8-tetramethyl-5,6,7,8-tetrahydro-naphthalen-2-yl)-1H-benzoimidazole-5-carboxylic acid, a retinoid derivative showing MIC value of 50 μg/mL against Candida krusei and Candida albicans. This sketch was drawn in ChemDraw^®^ Ultra 7.0. Perkin Elmer Italia S.p.A.

**Figure 2 pharmaceuticals-14-00962-f002:**
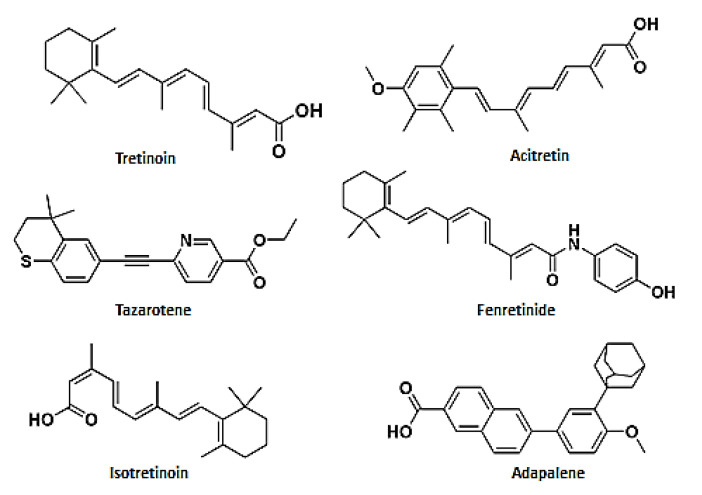
Chemical structures of the active retinoids against mycosis. The sketches were drawn in ChemDraw ^®^Ultra 7.0. Perkin Elmer Italia S.p.A.

**Table 2 pharmaceuticals-14-00962-t002:** Evaluation of retinoid efficacy against mycosis.

	Tretinoin	Tazarotene	Isotretinoin	Acitretin	Fenretinide	Adapalene
In vitro studiies	*A. fumigatus*	*C. glabrata*	*A. fumigatus*			
	*C. albicans*	*C. albicans*	*A. niger*			
	*Microsporum* spp.	*T. verrucosum*	*C. albicans*			
	*Trichophyton* spp.					
	*Epidermophyton* spp.					
In vivo studies	*M. furfur*	*A. niger*	*M. furfur*	*F. monophora*	*A. fumigatus*	*M. furfur*
	*A. fumigatus*	*T. rubrum*	*P. jiroveci*	*R. mucilaginosa*		
		*T. tonsurans*				
		*T. mentagrophytes*				
		*E. floccosum*				
		*C. albicans*				
		*A. flavus*				

## Data Availability

Data sharing not applicable.

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
