# Peer review of "Retinoids in Fungal Infections: From Bench to Bedside"

_pharmaceuticals, 2021, doi:10.3390/ph14100962_

Round 1
Reviewer 1 Report
Evaluation and comments to the manuscript ID pharmaceuticals-1376482, and entitled “Retinoids in Fungal infections: from bench to beside”.
Authors: Terenzio Cosio et al.
Overall, the manuscript is interesting and worth publishing.I have no comments as to the content of the manuscript, however, the aesthetics and technical aspects are very worrying.Therefore, the work must be significantly improved in this matter before it is accepted.
Below are some examples of remarks:
1) why is "spp." written in italics
2) please check and correct the affiliation record, eg. “Italy. elena.campione@uniroma2.it .”. Why the "dot" behind of Italy and the emil? Etc.
3) Keywords – Latin names of fungi (Candida, Aspergillus, Malassezia) are written in italics
4) ln 84: something wrong with the type of font in the word “acute promyelocytic leukemia”
5) Fgure 1: poor quality and something wrong with the notation of the reference literature
6) ln 202: “...R. Mucilaginosa...” – the second part with a lowercase letter
7) ln 206: “...Idris et al....” - it should be written: Idris et al. [23]. Please check the entire manuscript carefully regarding this note !!!
8) and many others,
9) I also ask the authors to think about the improvement of the title. Although I do not insist and leave the decisions on this matter to the author.
Author Response
Overall, the manuscript is interesting and worth publishing. I have no comments as to the content of the manuscript, however, the aesthetics and technical aspects are very worrying. Therefore, the work must be significantly improved in this matter before it is accepted.
- why is "spp." written in italics
Dear reviewer, thank you for the suggestion. Grammatical and formal mistaken have been corrected.
- please check and correct the affiliation record, eg. “Italy. elena.campione@uniroma2.it .”. Why the "dot" behind of Italy and the emil? Etc.
Dear reviewer, thank you for the suggestion. Grammatical and formal mistakens have been corrected.
- Keywords – Latin names of fungi (Candida, Aspergillus, Malassezia) are written in italics
Dear reviewer, thank you for the suggestion. Grammatical and formal mistakens have been corrected.
- ln 84: something wrong with the type of font in the word “acute promyelocytic leukemia”
Dear reviewer, thank you for the suggestion. Grammatical and formal mistakens have been corrected.
- Fgure 1: poor quality and something wrong with the notation of the reference literature
Dear reviewer, thank you for the suggestion. All figures in the main file have been modified and improve in quality and in reference.
6) ln 202: “...R. Mucilaginosa...” – the second part with a lowercase letter
Dear reviewer, thank you for the suggestion. Grammatical and formal mistakens have been corrected.
7)ln 206: “...Idris et al....” - it should be written: Idris et al. [23]. Please check the entire manuscript carefully regarding this note !!!
Dear reviewer, thank you for the suggestion. Grammatical and formal mistaken have been corrected.
8)and many others,
Dear reviewer, thank you for the suggestion. Grammatical and formal mistakens have been corrected. Moreover, the text has been edited.
9) I also ask the authors to think about the improvement of the title. Although I do not insist and leave the decisions on this matter to the author.
9)Dear Reviewer, we have decided to not change the title to underline the translational view of our article.
Reviewer 2 Report
The manuscript (Pharmaceuticals-1376482) of Terenzio Cosio, Roberta Gaziano, Guandalina Zuccardi, Gaetana Costanza, Sandro Grelli, Paolo di Francesco, Luca Bianchi and Elena Campione titled “Retinoids in fungal infections: from bench to bedside ” before publication in Pharmaceuticals
needs some minor corrections.
Comments:
Line 15 should be “dermatology, oncohematology” instead of “Dermatology, Oncohematology”
Line 22 should be “in vitro and in vivo” instead of “in vitro and in vivo” (in vitro - sometimes it is in italics, other times not. Should be in italics.
Line 41 should be “e.g.” instead of “eg”
Line 66 should be “anti-microbial” instead of “anti- microbial”
Line 146 should be “multidrug resistance” instead of “Multidrug resistance”
Table 1. The table is not readable and contains ambiguities,
for example, it is not stated in what solvent with a volume
of 3 mL is 1 g of 0.1% Tazarotene gel dissolved and what does “1mL1” mean?
In the whole table there should be “culture” instead of “colture”, and
should be “20 mg/kg” instead of “20 mg/Kg”.
Line 160 should be “-COOH“ instead of “COOH“,
should be “50 μg/mL“ instead of “50μg/mL“
Figures 1 and 2 contain low quality chemical formulas, these figures is not legible. It should be changed.
The article presents an interesting approach to the subject of retinoids, i.e. a different view of retinoids in the context of their antifungal activity. The presented work is interesting and thoroughly prepared.
Author Response
The manuscript (Pharmaceuticals-1376482) of Terenzio Cosio, Roberta Gaziano, Guandalina Zuccardi, Gaetana Costanza, Sandro Grelli, Paolo di Francesco, Luca Bianchi and Elena Campione titled “Retinoids in fungal infections: from bench to bedside” before publication in Pharmaceuticals needs some minor corrections.
Comments:
- Line 15 should be “dermatology, oncohematology” instead of “Dermatology, Oncohematology”
- Dear reviewer, thank you for the suggestions. All corrections have been made.
- Line 22 should be “in vitro and in vivo” instead of “in vitro and in vivo” (in vitro - sometimes it is in italics, other times not. Should be in italics.
2)Dear reviewer, thank you for the suggestions. All corrections have been made.
3)Line 41 should be “e.g.” instead of “eg”
3)Dear reviewer, thank you for the suggestions. All corrections have been made.
4)Line 66 should be “anti-microbial” instead of “anti- microbial”
4)Dear reviewer, thank you for the suggestions. All corrections have been made.
5)Line 146 should be “multidrug resistance” instead of “Multidrug resistance”
5)Dear reviewer, thank you for the suggestions. All corrections have been made.
6)Table 1. The table is not readable and contains ambiguities, for example, it is not stated in what solvent with a volume of 3 mL is 1 g of 0.1% Tazarotene gel dissolved and what does “1mL1” mean?
6)Dear reviewer, thank you for the suggestions. All corrections have been made in order to clarify any doubts.
7)In the whole table there should be “culture” instead of “colture”, and should be “20 mg/kg” instead of “20 mg/Kg”.
7)Dear reviewer, thank you for the suggestions. All corrections have been made.
8)Line 160 should be “-COOH“ instead of “COOH“, should be “50 μg/mL“ instead of “50μg/mL“
8)Dear reviewer, thank you for the suggestions. All corrections have been made.
9)Figures 1 and 2 contain low quality chemical formulas, these figures is not legible. It should be changed.
9)Dear reviewer, thank you for the suggestions. All figures have been edited and improved to made them more readable. Moreover, figure 2 has been divided in table and figure.
10)The article presents an interesting approach to the subject of retinoids, i.e. a different view of retinoids in the context of their antifungal activity. The presented work is interesting and thoroughly prepared.
10)Dear reviewer, thank you for your positive comment and for your interest. we are also grateful that the work has been understood and that its rationale can be perceived. Moreover, thank you for the accuracy of in-text notes and corrections, that improve the manuscript’s quality.
Reviewer 3 Report
Critique
Retinoids in fungal infections” from bench to bedside
T.Casio, R. Gaziano. G. Zuccardi, et al.
Comments:
This is an excellent and relatively comprehensive review of retinoid compounds and their ability to treat fungal infections. The appropriate references appear to be cited, although several are clinical observations. This points out the need, as the authors express in their conclusions, for well controlled clinical trials. Particularly interesting was the use of new delivery vehicles to deal with some of the problems of the retinoid derivatives. This work will be of interest to infectious disease clinicians as well as scientists interested in the chemistry of retinoids and their derivatives.
However, the English in the manuscript needs to be improved. I suggest having a native English speaker edit this manuscript prior to publication
Author Response
This is an excellent and relatively comprehensive review of retinoid compounds and their ability to treat fungal infections. The appropriate references appear to be cited, although several are clinical observations. This points out the need, as the authors express in their conclusions, for well controlled clinical trials. Particularly interesting was the use of new delivery vehicles to deal with some of the problems of the retinoid derivatives. This work will be of interest to infectious disease clinicians as well as scientists interested in the chemistry of retinoids and their derivatives.
Kind reviewer, first of all thank you for the review of the work and the positive notes on the quality of our manuscript. In the future, new trials will certainly be needed and above all the setting up of phase I trials to evaluate new formulations. In addition to your review, we are available for future collaborations given your interest.
However, the English in the manuscript needs to be improved. I suggest having a native English speaker edit this manuscript prior to publication
Dear Reviewer, the manuscript was edited by a native speaker colleague in order to improve the syntax and make it more readable.
Round 2
Reviewer 1 Report
The current version of the manuscript is significantly better compared to the original version. Therefore, despite minor flaws, I believe that the work can be accepted for publication.